# Direct observations of energy transfer from resonant electrons to whistler-mode waves in magnetosheath of Earth

N. Kitamura ®[1,2] ✉, T. Amano ®[2], Y. Omura ®[3], S. A. Boardsen ®[4,5], D. J. Gershman[4], Y. Miyoshi ®[1], M. Kitahara[6], Y. Katoh ®[6], H. Kojima ®[3], S. Nakamura[1], M. Shoji ®[1], Y. Saito[7], S. Yokota ®[8], B. L. Giles[4], W. R. Paterson[4], C. J. Pollock[9], A. C. Barrie[4,10], D. G. Skeberdis[4,11], S. Kreisler[4,10], O. Le Contel ®[12], C. T. Russell[13], R. J. Strangeway ®[13], P.-A. Lindqvist[14], R. E. Ergun[15], R. B. Torbert ®[16,17] & J. L. Burch ®[17]

Electromagnetic whistler-mode waves in space plasmas play critical roles in collisionless energy transfer between the electrons and the electromagnetic field. Although resonant interactions have been considered as the likely generation process of the waves, observational identification has been extremely difficult due to the short time scale of resonant electron dynamics. Here we show strong nongyrotropy, which rotate with the wave, of cyclotron resonant electrons as direct evidence for the locally ongoing secular energy transfer from the resonant electrons to the whistler-mode waves using ultra-high temporal resolution data obtained by NASA's Magnetospheric Multiscale (MMS) mission in the magnetosheath. The nongyrotropic electrons carry a resonant current, which is the energy source of the wave as predicted by the nonlinear wave growth theory. This result proves the nonlinear wave growth theory, and furthermore demonstrates that the degree of nongyrotropy, which cannot be predicted even by that nonlinear theory, can be studied by observations.

The interaction between electromagnetic fields and charged particles is central to collisionless plasma dynamics in space. Right-hand polarized whistler-mode waves have been the subject of many studies owing to their efficient pitch-angle scattering[1] and acceleration of electrons[2–4] and play important roles in the solar wind[5,6], in collisionless shock waves[7–9], and in planetary magnetospheres (creation of electron radiation belts and diffuse aurora)[10–20]. Whistler-mode waves are linearly unstable, for instance, in the presence of electron temperature anisotropy (higher temperature perpendicular to the magnetic field)[3,4,10]. Observations of linearly unstable velocity distribution functions have been considered as evidence for the wave growth[21–23]. Quasi-linear theory has been widely used to predict how electrons interacting with incoherent waves diffuse in phase space[11,13,16,18].

On the other hand, nearly-monochromatic right-hand circularly polarized waves, which must be coherent, are often observed in space[24–27]. Such coherent waves are expected to lead to much more efficient wave-particle interaction owing to the ability of phase trapping of resonant particles within a wave potential, which causes non-diffusive particle transport in phase space[14,19,28–31]. The nonlinear theory for an inhomogeneous medium[14,28] predicts individual particle trajectories in phase space, especially the occurrence of such trapping, depending on gradients of the magnetic field intensity and the plasma density in addition to other parameters provided by in situ observations. If the flux of trapped particles differs from that of untrapped particles, a resonant current is formed, and the resonant current plays the dominant role in the nonlinear wave-particle interaction. The magnitude of the resonant current seen as nongyrotropy of particles,

however, cannot be predicted because it is affected by the accumulated history of the interaction between the resonant electrons and the waves at different locations. Although nongyrotropy of protons resonantly interacting with much lower (a factor of about 1000) frequency electromagnetic ion cyclotron waves have been detected recently[32–36], electron nongyrotropy is prohibitively difficult to identify since the wave frequencies are much higher than the temporal resolution of particle instruments.

Here, we show strongly nongyrotropic electron velocity distribution functions (VDFs) rotating with whistler-mode waves around the cyclotron resonance velocity as smoking-gun evidence for locally ongoing energy supply to the wave by analyzing data obtained by the Magnetospheric Multiscale (MMS) spacecraft[37]. We compare the observed features with the nonlinear wave-particle interaction theory for coherent waves, and find good agreement.

## Results

### Dataset for electromagnetic fields
The magnetic field measured by the fluxgate magnetometers (FGM)[38] (burst data, 128 samples s$^{-1}$) were used as the background magnetic field ($\mathbf{B}_0$). Since the different instruments have different temporal resolutions as described below, $\mathbf{B}_0$ was linearly interpolated and used to determine the field-aligned coordinate (FAC) system for each measurement. The +$z$ direction in FAC was defined to be the direction of $\mathbf{B}_0$. The +$y$ direction was defined as the cross product of the +$z$ direction and the vector pointing to the Sun from the Earth. The +$x$

direction was defined to complete an orthogonal right-handed coordinate system.

To obtain electromagnetic fields of the whistler-mode waves, we analyzed the burst data obtained by the search-coil magnetometers (SCM)[39] (8192 samples s$^{-1}$) and electric field double probes (EDP)[40,41] (8192 samples s$^{-1}$). Except for overview plots, data from 15:59:08 to 15:59:24 Universal Time (UT) (16 s) on 25 December 2016 (Event 1) and from 05:26:21 to 05:26:29 UT (8 s) on 28 December 2016 (Event 2) were used. After the coordinate transformation to FAC, a fast Fourier transform (FFT) and an inverse FFT were applied to the SCM and EDP data to retrieve waveform data ($\mathbf{B}_w$ and $\mathbf{E}_w$) for whistler-mode waves that include the components between 70 and 400 Hz (Event 1) or 10 and 300 Hz (Event 2)[42]. We defined the wave FAC (wFAC) system using the $x$ and $y$ components of $\mathbf{B}_w$ (perpendicular to $\mathbf{B}_0$) in FAC. The +$z$ direction in wFAC is the same as that in FAC (direction of $\mathbf{B}_0$). The +$x$ direction was defined as the direction of $(\mathbf{B}_w)_{xy}$. The +$y$ direction was defined to complete an orthogonal right-handed coordinate system.

### Event 1 overview
The interval focused on hereafter as Event 1 (around 15:59:19 UT on 25 December 2016) was near the postnoon (magnetic local time: about 13.6 h) magnetopause, and was probably close to the magnetosheath-side separatrix of the magnetopause reconnection that occurred northward of MMS (Figs. 1 and 2). The MMS spacecraft crossed the magnetopause from the magnetosphere to the magnetosheath. The characteristics of the magnetosphere are northward-directed $\mathbf{B}_0$ (+$z$ in

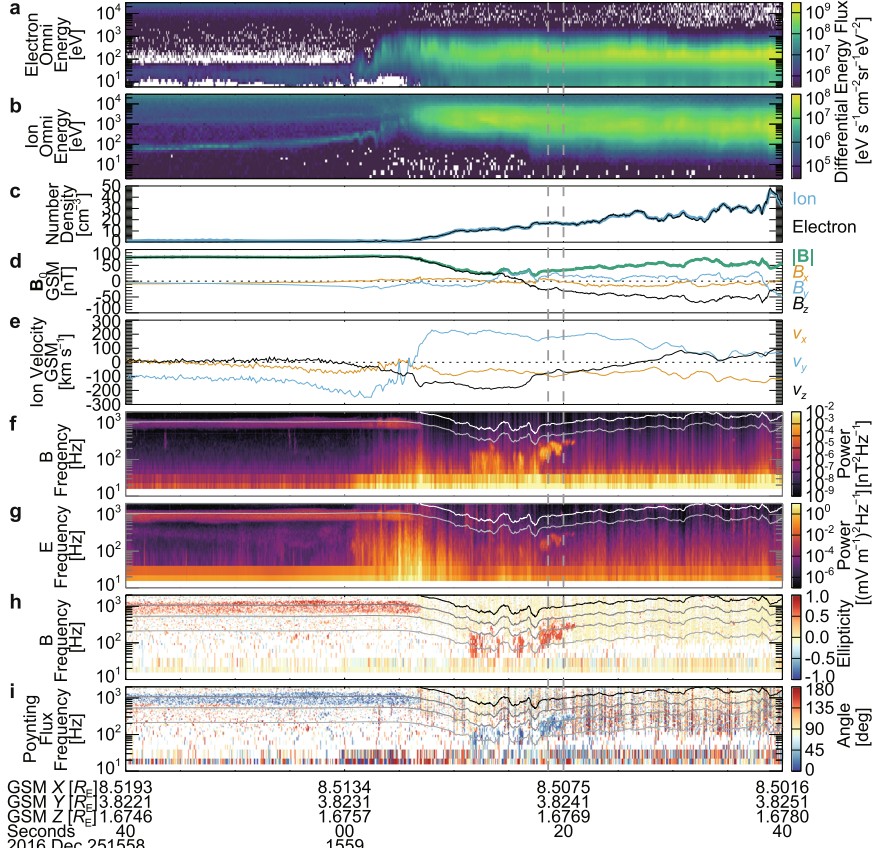

**Fig. 1 | Overview of MMS1 burst data (Event 1). a** Omni-directional energy spectrum of electrons (photoelectrons were subtracted). **b** Omni-directional energy spectrum of ions. **c** Number density of ions (light blue) and electrons (black). **d, e** the background magnetic field ($\mathbf{B}_0$) and ion bulk velocity in the geocentric solar magnetic (GSM) coordinates. **f, g** Wave power spectrum of magnetic and electric fields with the cyclotron resonance velocity ($f_{ce}$) (white) and $0.5f_{ce}$ (grey). **h, i** Spectra of ellipticity and angle of Poynting flux from $\mathbf{B}_0$ with $f_{ce}$ (black), $0.5f_{ce}$

(dark grey), $0.25f_{ce}$ (grey), and $0.1f_{ce}$ (light grey). Enhancements of electromagnetic right-hand polarized (positive ellipticity) waves corresponds to whistler-mode waves. Vertical grey dashed lines indicate the interval shown in Fig. 4. Spacecraft positions in Earth radii ($R_E$) are shown at the bottom. MMS crossed the magnetopause southward of the reconnection site from the magnetosphere to the magnetosheath (see also Fig. 2).

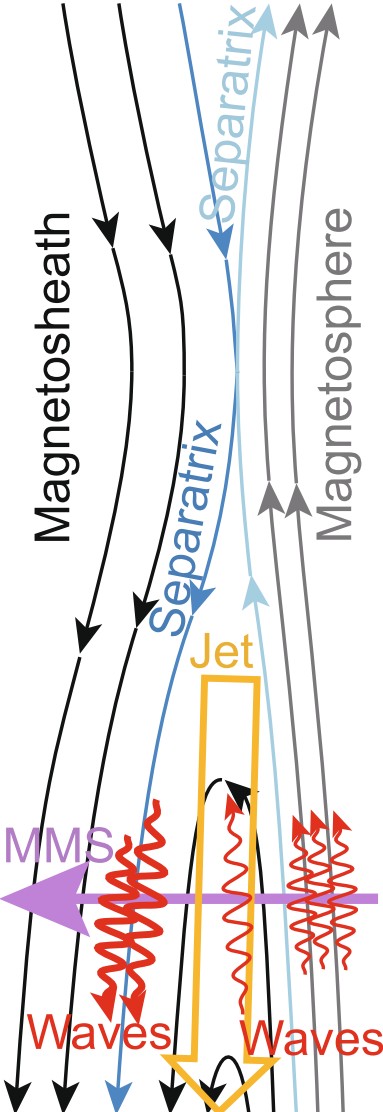

**Fig. 2 | Schematic of the magnetopause crossing and whistler-mode waves (Event 1).** MMS crossed the magnetopause southward of the reconnection site from the terrestrial magnetosphere to the magnetosheath. The interval with whistler mode waves focused as Event 1 (around 15:59:19 UT on 25 December 2016) was close to the magnetosheath-side separatrix (blue curve) of the magnetopause reconnection that occurred northward of MMS.

the geocentric solar magnetic (GSM) coordinates) and the existence of hot electrons and ions (>10 keV), which were observed by the fast plasma investigation (FPI)[43] (see Methods, subsection Electron and ion measurements by FPI), while the magnetosheath is characterized by a high-density warm plasma and $\mathbf{B}_0$ directed southward (Fig. 1a–d). Just before and during the rotation of $\mathbf{B}_0$, the GSM-$z$ component of ion bulk velocity reached up to −200 km s$^{-1}$, which is a typical feature of the magnetopause reconnection (southward directed jet)[44,45] (Fig. 1e). The wave power of SCM and EDP data is enhanced mainly below $0.5f_{ce}$ at the various locations around the reconnection jet (Fig. 1e–g), where $f_{ce}$ is the electron cyclotron frequency. Some wave enhancements were right-hand polarized (positive ellipticity) electromagnetic whistler-mode waves propagating parallel to $\mathbf{B}_0$ (angle of Poynting flux from $\mathbf{B}_0$ close to 0°) (Fig. 1f–i) (see Methods, subsection Analysis related to wave spectra (power, ellipticity, and Poynting flux angle)). Because the separation of the spacecraft (<11 km) was about 10 times the gyro-radius of nongyrotropic electrons discussed later (Supplementary Fig. 1), observational differences between the spacecraft cannot be

seen over this time scale. Observed features are schematically summarized in Fig. 2. Appearance of whistler-mode waves in the jet and around the separatrices itself have been reported by many studies[22,46–50]. Although we focus on the whistler-mode waves around the magnetosheath-side separatrix[22,46,47] hereafter, they have not been reported as frequently as those around the magnetosphere-side separatrix[22,47–50].

## Estimate of the dispersion relation and resonance velocity
Under the cold plasma approximation (CPA), the wave angular frequency ($\omega$) and wavenumber ($k$) of whistler-mode waves that propagate along $\mathbf{B}_0$ satisfy the dispersion relation given as,

$$c^2 k^2 = \omega^2 + \frac{\omega \omega_{pe}^2}{\Omega_{ce} - \omega} \tag{1}$$

where $c$, $\omega_{pe}$, and $\Omega_{ce}$ are the speed of light, the electron plasma frequency, and the electron cyclotron angular frequency, respectively. Here, $\omega_{pe} = \sqrt{q^2 n_p / m_e \varepsilon_0}$ and $\Omega_{ce} = |q| B_0 / m_e$, where $q$, $n_p$, $m_e$, $B_0$, and $\varepsilon_0$ are the electric charge (negative for electrons), the plasma density, the mass of electrons, the intensity of $\mathbf{B}_0$, and the permittivity of vacuum, respectively. The wave frequency ($f = \omega / 2\pi$), which was calculated from the rotation period of $\mathbf{B}_w$ (see Methods, subsection Calculation of wave frequency), was about 220 Hz (Supplementary Fig. 2a). According to Eq. (1) with an ion density and a magnetic field intensity observed by MMS1 (16.9 cm$^{-3}$ and 33.1 nT, average between 15:59:19.509 and 15:59:19.779 UT) for $n_p$ and $B_0$, $k$ becomes about 0.432 rad km$^{-1}$. The nonrelativistic cyclotron resonance velocity ($V_{res} = (\omega - \Omega_{ce})/k$) is about 10,300 km s$^{-1}$ (minimum resonant energy: about 300 eV), which corresponds to an energy of 500 eV for an electron with a pitch angle (PA) of 141°. We check the validity of $k$ derived under CPA in Methods (subsection Validation of estimated $k$) using the phase difference of $\mathbf{B}_w$ between MMS1 and MMS4.

## Electron distribution function
The electron VDF exhibited a power-law decrease with increasing energy above about 100 eV, and the start of the decrease depends on PA (Fig. 3a, Supplementary Fig. 3a for detail). A gradient of phase space density (PSD) around $V_{res}$ adequate for the initial linear growth of whistler-mode waves (increasing toward a PA of 90°) is found only at PAs larger than about 130° at energies higher than about 200 eV (Fig. 3a, Supplementary Fig. 3b for detail).

We investigate the existence of nongyrotropic electrons using the relative phase angle ($\zeta$) versus PSD histograms, where $\zeta$ is the angle between the direction of $\mathbf{B}_w$ in FAC $x$-$y$ plane (+$x$ direction in wFAC) and the direction of the electron velocity. The +$y$ direction in wFAC corresponds to $\zeta = 90°$. The burst data obtained by the FPI-dual electron spectrometer (DES) are disassembled into 128 groupings, each of 128 simultaneous measurements (integration time: 196 μs). This disassembling made it possible to establish the ultra-high time resolution that can resolve variations of electron fluxes within a wave period about 5 ms. Using the disassembled FPI-DES data (time and look directions) and $\mathbf{B}_w$ from each spacecraft, we calculated $\zeta$ for each measurement and constructed combined electron VDFs using all four spacecraft (see Methods, subsection Electron and ion measurements by FPI). The histograms (Fig. 3b–d) clearly show that the electrons around $V_{res}$ exhibit strong nongyrotropy and had a broad dip of PSD (about 40% decrease from the peak in the most prominent case) around $\zeta$ of about 90°. Note that this nongyrotropy is seen in a system that rotates with the whistler-mode wave, and thus it is essentially different from the nongyrotropy in a constant orientation with respect to the current layer, which is observed at the electron dissipation region of magnetic reconnection[51].

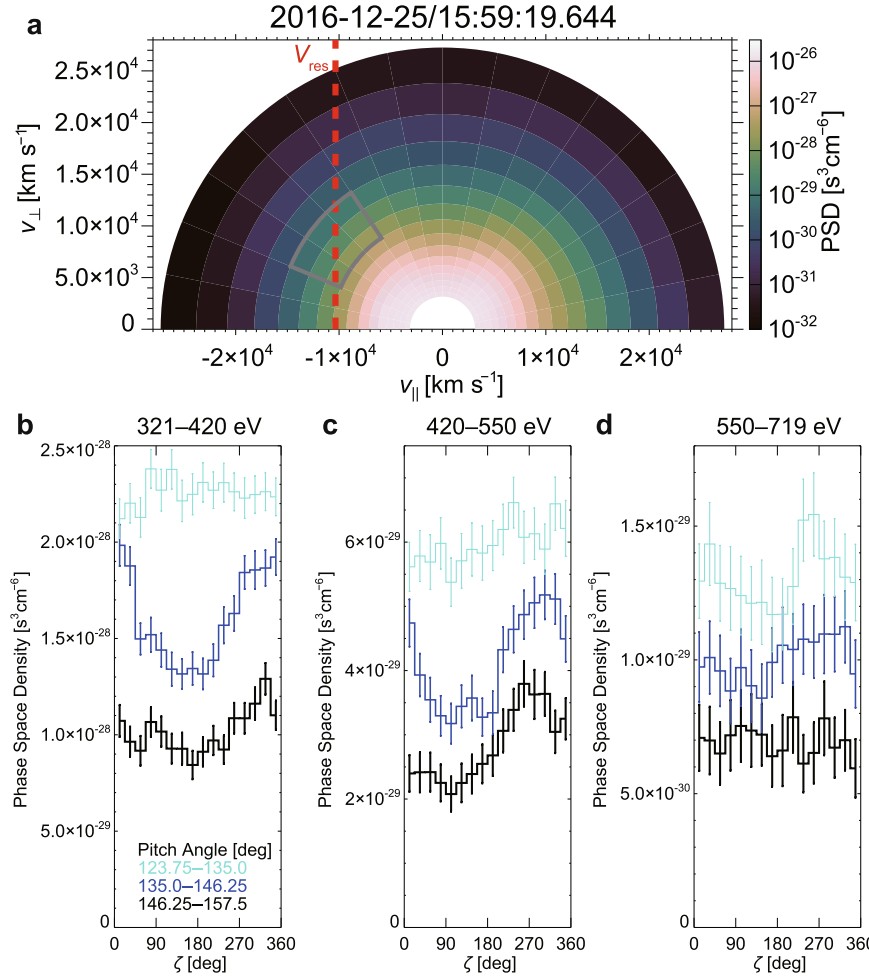

**Fig. 3 | Electron velocity distribution function (Event 1). a** Gyro-averaged electron velocity distribution function (28.3–2112 eV) with the cyclotron resonance velocity ($V_{res}$) shown as a red dashed line. **b–d** histograms of electron phase space density (PSD) in $\zeta$ direction at 3 energy bins (fan-shaped area surrounded by a gray curve in Fig. 3a). The nongyrotropy significantly exceeded 2σ error bars (see Methods, subsection Electron and ion measurements by FPI) around $V_{res}$, while the electrons did not exhibit clear nongyrotropy around the pitch angle of about 130° below about 550 eV. This indicates confinement of this electron nongyrotropy around $V_{res}$. Electron data from 9 temporal bins (270 ms) around 15:59:19.644 UT (Fig. 4) from each spacecraft were used (see Methods, subsection Electron and ion measurements by FPI). The fan-shaped area surrounded by a gray curve corresponds to the pitch angle and energy ranges for the calculation of the resonant current ($\mathbf{J}_{res}$) for Event 1.

The $\zeta$ versus time ($\zeta$-$t$) spectra of normalized electron fluxes, in which the differential electron fluxes for the $\zeta$-$t$ spectra were normalized by the averaged fluxes of all $\zeta$ bins to focus on nongyrotropy, indicate that the dip of PSD became clearest around 15:59:19.0 and 15:59:19.6 UT (Fig. 4). The $y$ component in wFAC of the resonant current ($\mathbf{J}_{res}$) (see Methods, subsection Electron and ion measurements by FPI), which is driven by the nongyrotropic resonant electrons, was continuously positive from 15:59:18.8–15:59:19.8 UT. Because the +$y$ direction in wFAC corresponds to −$\mathbf{E}_w$ in the FAC $x$-$y$ plane for a wave propagating parallel to $\mathbf{B}_0$, $\mathbf{J}_{res} \cdot \mathbf{E}_w < 0$, which indicates secular energy transfer from the resonant electrons to the wave. The energy transfer rate becomes about 5 pW m$^{-3}$, if about 4 nA m$^{-2}$ and about 1.3 mV m$^{-1}$ are used as the typical magnitudes (around 15:59:19.6 UT) of the $y$ component in wFAC of $\mathbf{J}_{res}$ and $\mathbf{E}_w$, respectively (Fig. 4a, h).

### Electron measurements by the electron drift instruments

Independent evidence is provided by one of the electron drift instruments (EDIs)[52] on MMS2, which continuously measures 500 eV electrons with PAs larger than about 140° (see Methods, subsection Electron measurements by EDI). One of the channels that measured close to $V_{res}$ (PA of about 140° at 500 eV) detected strong modulation of the electron flux, and the dips were almost always around $\zeta = 90°$

when the wave amplitude became large (Fig. 5). This observation is in strong agreement with the results obtained from the analysis of DES data. The appearance of the dip around $\zeta = 90°$ roughly agrees with the characteristic identified in the simulations of nonlinear growth of whistler-mode waves[53–57]. Around both of the two intervals when the nongyrotropy became strongest, the modulation of electron fluxes became rapidly weaker with increasing PA (Fig. 4b–g). This weakening and the pitch-angle dependence of $\zeta$-$t$ spectra of normalized electron fluxes demonstrate that the nongyrotropy was confined to the vicinity of $V_{res}$.

### Event 2 in the magnetosheath

Further evidence for the nonlinear wave-particle interaction is provided by the second event (Event 2) in a trough of magnetic field intensity in the magnetosheath region (Supplementary Fig. 4). In the trough of magnetic field intensity, the decrease in magnetic pressure was compensated with an increase in plasma (mainly ion) pressure (Supplementary Fig. 5). Enhancements of whistler-mode waves in such structures have been called lion roars and are a well-known feature in the magnetosheath[24–26,28,42,58]. Because the separation of the spacecraft (<12 km) was about 3 times the gyro-radius of nongyrotropic electrons at 500 eV discussed later (Supplementary Fig. 6), observational

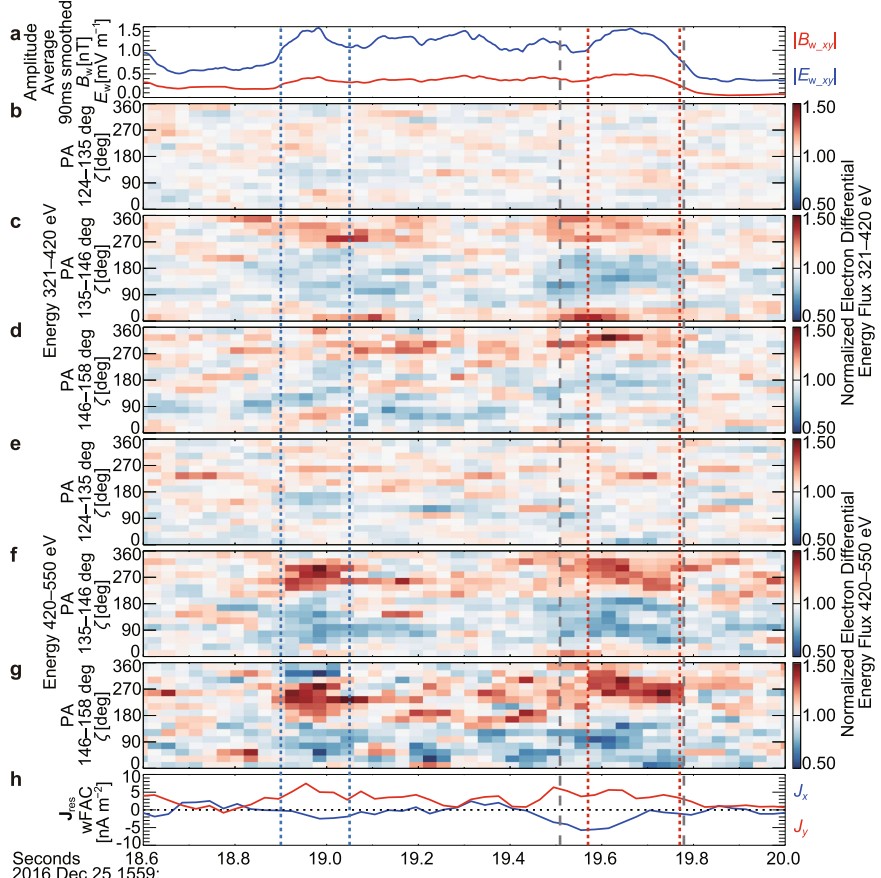

**Fig. 4 | Temporal variation of nongyrotropic electrons at multiple energy and pitch angle bins and the resonant current (Event 1). a** Amplitude of wave component of the magnetic field ($B_w$) and the electric field ($B_w$) in the field-aligned coordinate (FAC) x-y plane. Moving-average values from all four spacecraft measurements with a window of 90 ms, which is comparable with the window for the analysis of electron data (3 temporal bins (see Methods, subsection Electron and ion measurements by FPI)), are shown. **b–g** $\zeta$-t spectra of normalized differential energy fluxes of electrons (energy bin: 320.9–419.7 or 419.7–549.8 eV and pitch angle (PA) bin: 123.75°–135.0°, 135.0°–146.25°, or 146.25°–157.5°). **h** the resonant current ($J_{res}$) (x and y components in the wave field-aligned coordinate (wFAC)). Vertical gray dashed lines indicate the interval analyzed for Fig. 3. Vertical blue dotted and red dotted lines indicate the intervals shown in Fig. 5.

differences between the spacecraft cannot be clearly seen over these time scales.

Similar to Event 1, clear nongyrotropy was detected at multiple energy and/or pitch angle bins by DES and EDI in the wave packet around 05:26:23.7 UT (Figs. 6 and 7), which had an almost constant $f$ of about 23 Hz (Supplementary Fig. 7a). According to Eq. (1) with an ion density and a magnetic field intensity observed by MMS1 (12.3 cm$^{-3}$ and 11.6 nT, average between 05:26:23.549 and 05:26:23.819 UT) for $n_p$ and $B_0$, $k$ and $V_{res}$ become about 0.183 rad km$^{-1}$ and about 10,350 km s$^{-1}$ (minimum resonant energy: about 300 eV), which corresponds to an energy of 500 eV for an electron with a PA of 141°. Although the electron PSD at about 500 eV was lower than Event 1 (Supplementary Figs. 8 and 9a), the nongyrotropy with a dip at $\zeta$ around 90°, which corresponds to $J_{res}$ for $J_{res} \cdot E_w < 0$, was significant at multiple energy and/or pitch angle bins around $V_{res}$ (Supplementary Fig. 9). The energy transfer rate becomes about 1.5 pW m$^{-3}$, if about 5 nA m$^{-2}$ and about 0.3 mV m$^{-1}$ are used as the typical magnitudes (around 05:26:23.7 UT) of the y component in wFAC of $J_{res}$ and $E_w$, respectively (Fig. 6a, h).

## Discussion

The observed waves must be coherent at least in each wave packet because $f$ was quite stable except at the boundary of the wave packets where discontinuity in wave phase between the wave packets can often prevent accurate measurements of the rotation period (Supplementary Figs. 2a, b, 7a, b). The amplitude variation of the wave packets leads to broadening of wave power in frequency (Fig. 1f, g). The

waveform around the time intervals when the electrons exhibit strong nongyrotropy (Figs. 5f, 7f) is apparently far from incoherent noise, which is assumed in the linear growth theory[2–4], but close to sinusoidal. Thus, we compare the observed features with the nonlinear wave-particle interaction theory for coherent waves[28].

The theory shows a crucial role of the phase trapping of particles, which leads to efficient nonlinear wave growth and particle acceleration, in a coherent wave with temporal variation in the wave frequency and/or spatial variation in the background magnetic field intensity. The temporal and/or spatial inhomogeneity is characterized by a parameter $S$ called an inhomogeneity factor which must be within the range from −1 to 0 to cause phase trapping. Although the nonlinear theory so far has been used mainly for whistler-mode waves in the magnetosphere[14,59], the process is fundamental and similar interactions can occur in any region of space as far as the condition for the occurrence of the phase trapping is satisfied. The equations to calculate $S$ and trapping frequency ($\omega_{tr}$), which are used to estimate the width of trapping, are summarized by ref. 28. We have used the set of equations, including the relativistic effect for estimating $S$, but it is otherwise ignored (because the correction is only of minor importance for Event 1 and 2). Four-point observations by MMS allow us to obtain the gradient of $B_0$ (= |$\mathbf{B_0}$|) along $\mathbf{B_0}$ (see Methods, subsection Gradient of the background magnetic field intensity (grad $B_0$)) that is necessary to estimate $S$ from the observations. We used grad$B_0$ along $\mathbf{B_0}$ of about 20 pT km$^{-1}$ (Supplementary Fig. 2d) to derive the gradient of $\Omega_{ce}$, in addition to the parameters used for the estimation of dispersion

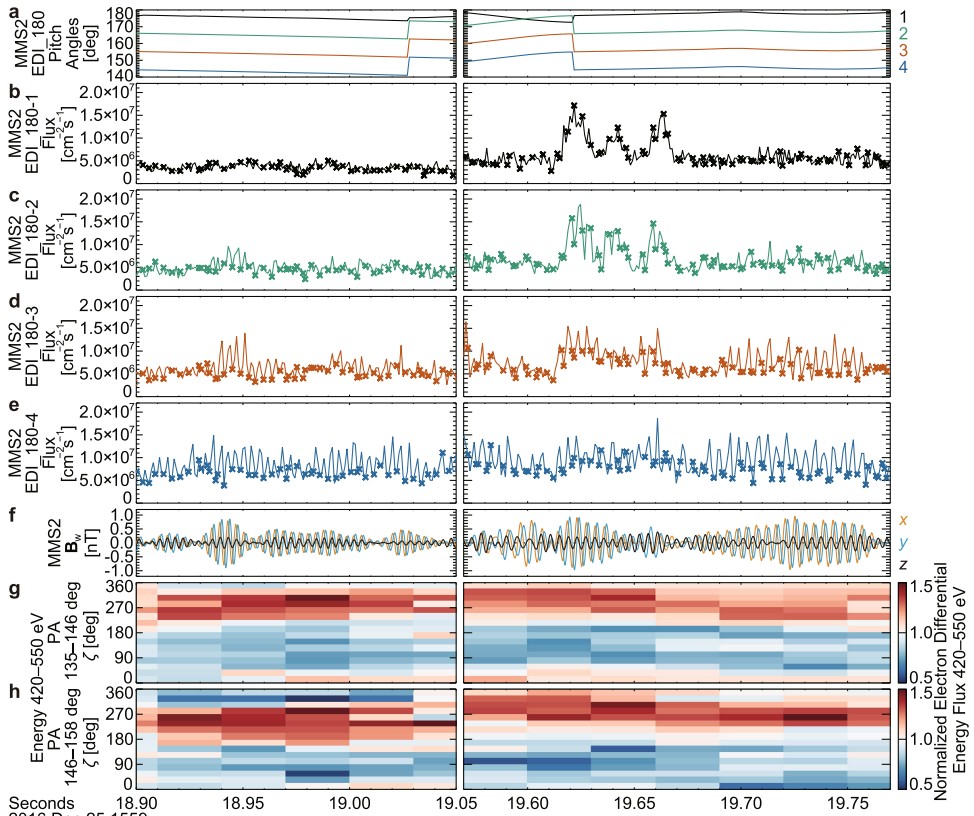

**Fig. 5 | Modulation of electron flux (Event 1). a** Pitch angle (PA) variation of looking directions of each channel of the electron drift instruments that observe the side anti-parallel to $\mathbf{B}_0$ (EDI_180). **b–e** Electron flux (500 eV) measured by 4 channels of EDI_180. The data points measured within 60° from $\zeta = 90°$ are highlighted with crosses. **f** Waveform of the magnetic field ($\mathbf{B}_w$) in the field aligned coordinate (FAC). **g, h** $\zeta$-$t$ spectra of normalized differential energy fluxes of electrons (energy bin: 419.7–549.8 eV and PA bin: 135.0°–146.25° or 146.25°–157.5°). Both measurements by Fast Plasma Investigation-Dual Electron Spectrometers and EDI_180 indicate the existence of a dip of electron fluxes at $\zeta$ around 90° around $V_{res}$. When $B_w$ was about 0.5 nT, electrons around 500 eV exhibited nongyrotropy in the PA range of around 135°–160° (also see Fig. 3c), this range is consistent with the width of trapping derived by a rough estimation.

relation for Event 1. Because there was no clear tendency of frequency variation (Supplementary Figs. 2a, 7a), we neglect the term related to the frequency variation. Thus, $S$ becomes proportional to grad $B_0$ along $\mathbf{B}_0$. In this calculation, a constant density along $\mathbf{B}_0$ was assumed. Estimated $S$ at $V_{res}$ with various $B_w$ and $v_{e\perp}$ for Event 1 are summarized in Table 1. This estimate indicates that $S$ satisfied the condition ($-1<S<0$) due to an appropriate magnitude of the gradient around the region in phase space where electrons became strongly nongyrotropic. For Event 2, the magnitude of grad $B_0$ along $\mathbf{B}_0$ was comparable with the accuracy of grad $B_0$ (see Methods, subsection Gradient of the background magnetic field intensity (grad $B_0$)) and thus, we briefly discuss the plausible range of $S$ only at about 500 eV ($v_{e\perp} = 8300$ km s$^{-1}$ at $V_{res}$). If 25 pT km$^{-1}$ (Supplementary Fig. 7d) was used as the upper limit and $B_w$ of 0.6 (0.3) nT, $S$ was larger than $-0.5$ ($-1.03$). This also indicates that $S$ was likely in the range between $-1$ and 0.

If $S$ is about $-0.4$, which is the optimum condition for nonlinear growth, the width of trapping potential in phase space becomes about $2\omega_{tr}/k$ (electrons within $\omega_{tr}/k$ from $V_{res}$ can be trapped)[14,28]. If $S$ is about 0, which is the condition that the width becomes largest, the width becomes about $4\omega_{tr}/k$. Around 500 eV ($v_{e\perp}$ of about 8500 km s$^{-1}$) and the wave amplitude ($B_w$) of about 0.5 nT, $\omega_{tr}/k$ becomes 1,300 km s$^{-1}$ for Event 1. At 500 eV, electrons within $\omega_{tr}/k$ ($2\omega_{tr}/k$) from $V_{res}$ correspond to those in the PA range of 133°–151° (125°–167°), if the values of $\omega_{tr}/k$ estimated above are used as a rough estimation. This size of the region in phase space where the nongyrotropy appeared in the vicinity of $V_{res}$ (Figs. 3–5) was consistent with the width of the trapping, which covered the PA range of about 135°–160° at

500 eV. For Event 2 ($B_w$ of about 0.6 nT), $\omega_{tr}/k$ becomes about 2191 km s$^{-1}$ around 500 eV ($v_{e\perp}$ of about 8,300 km s$^{-1}$) and within $\omega_{tr}/k$ from $V_{res}$ correspond to the PA range of 128°–161° at 500 eV. This is roughly consistent with the region in phase space where the nongyrotropy appeared (Fig. 7).

The growth rate is the critical parameter that indicates the growth or damping of waves and is directly connected to the nongyrotropy and the energy transfer rate. Although the presence of phase trapping can be predicted by the theory, the magnitude of the nongyrotropy cannot be predicted and must be assumed for calculations of theoretical nonlinear growth rates[14,28]. In the present study, because $\mathbf{J}_{res}$ was derived from the observational data, we can use it for the calculation of growth rate instead of such an assumption.

**Table 1 | Inhomogeneity factor (S) for nonlinear wave growth at $V_{res}$ (Event 1)**

| $B_w$ | $v_{e\perp}$ (energy, pitch angle) | | |
|---|---|---|---|
| | 5000 km s$^{-1}$ (373 eV, 154°) | 8500 km s$^{-1}$ (508 eV, 140°) | 12,000 km s$^{-1}$ (711 eV, 131°) |
| 0.2 nT | −0.75 | −0.49 | −0.40 |
| 0.5 nT | −0.30 | −0.20 | −0.16 |
| 0.8 nT | −0.19 | −0.12 | −0.10 |

Parameters observed by MMS1 around 15:59:19.64 UT were used for this calculation of $S$. At $V_{res}$ of −10,300 km s$^{-1}$, estimated $S$ satisfied the condition that the trapping can occur ($-1<S<0$) in the range of $v_{e\perp}$ where electrons became strongly nongyrotropic (Fig. 3).

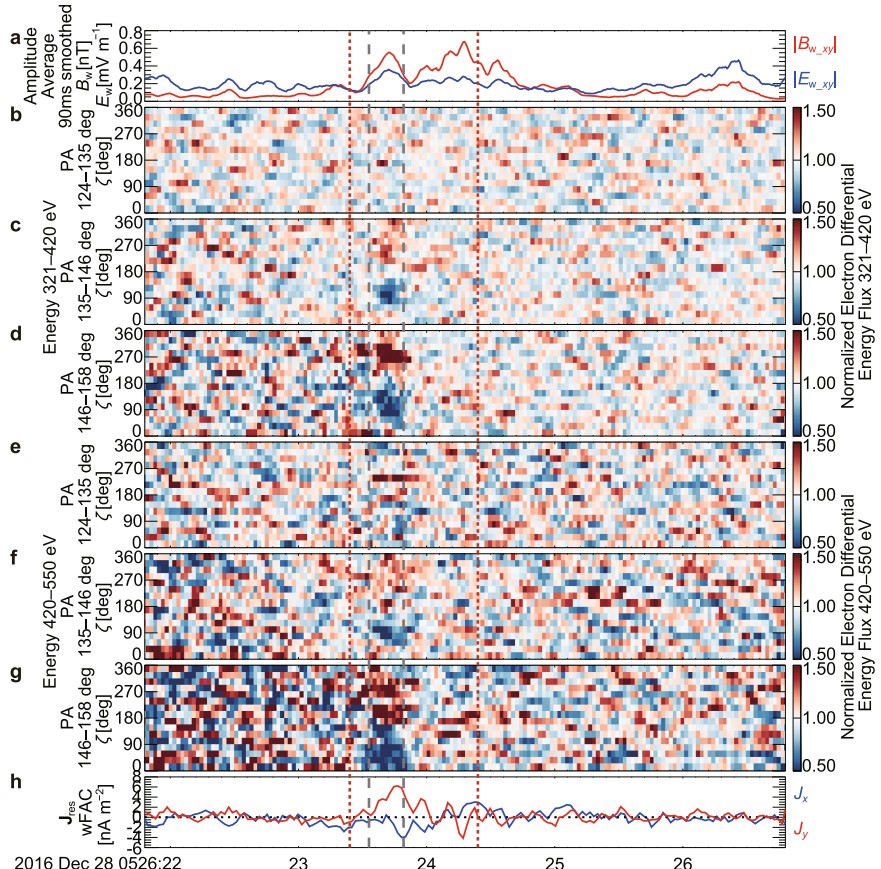

**Fig. 6 | Temporal variation of nongyrotropic electrons at multiple energy and pitch angle bins and the resonant current (Event 2).** The format is the same as that of Fig. 4. Vertical gray dashed lines indicate the interval analyzed for Supplementary Figs. 8 and 9. Vertical red dotted lines indicate the intervals shown in Fig. 7.

The nonlinear growth rate[28] can be estimated as,

$$\Gamma_N = -\frac{\mu_0 V_g}{2} \frac{J_E}{B_w},  \quad (2)$$

where $\mu_0$ is the magnetic permeability in vacuum, $V_g$ is the group velocity derived during the estimate of $S$, and $J_E$ corresponds to the $y$-component in wFAC of $-\mathbf{J}_{res}$. If about 0.4 nT, about 4900 km s$^{-1}$, and about 4 nA m$^{-2}$ are used as $B_w$ (Fig. 4a), $V_g$, and $J_E$ (Fig. 4h), respectively, $\Gamma_N$ becomes about 31 rad s$^{-1}$, which corresponds to about $0.02\omega$ and about $5 \times 10^{-3}\Omega_{ce}$ for Event 1. If about 0.5 nT, about 1500 km s$^{-1}$, and about 5 nA m$^{-2}$ are used as $B_w$ (Fig. 6a), $V_g$, and $J_E$, (Fig. 6h), respectively, $\Gamma_N$ becomes 9.4 rad s$^{-1}$, which corresponds to about $0.07\omega$ and about $5 \times 10^{-3}\Omega_{ce}$ for Event 2.

These two individual observations of strongly nongyrotropic electrons around $V_{res}$ during two events provide smoking-gun evidence of locally ongoing energy transfer from cyclotron resonant electrons to whistler-mode waves. With an appropriate magnitude of grad $B_0$ along $\mathbf{B}_0$, the condition became suitable for nonlinear wave growth due to phase trapping. The size of the region in phase space where the nongyrotropy appeared was consistent with the width of the theoretically expected phase trapping. Although the nonlinear wave growth due to phase trapping of electrons has been discussed exclusively for whistler-mode waves in the magnetosphere, identification of nongyrotropy has not been established there. The successful identification near the reconnection and in the magnetosheath indicates that the nonlinear wave growth may play a role in broader applications in space if the appropriate condition is satisfied.

## Methods

### Gradient of the background magnetic field intensity (grad $B_0$)

The FGM data from the other spacecraft were linearly interpolated in time to the MMS1 time tags for the calculations of grad $B_0$. Because the accuracy of FGM is ~0.1 nT[38] and the maximum separation of the MMS spacecraft along $\mathbf{B}_0$ was about 6 and about 8 km for Event 1 and 2 (Supplementary Figs. 1 and 6), it may become significant, when the magnitude of grad $B_0$ becomes larger than about 17 and about 12.5 pT km$^{-1}$, respectively. Because this calculation of grad $B_0$ was performed under the assumption that grad $B_0$ was flat in the tetrahedron of the four spacecraft, smaller-scale structures may cause an additional error of grad $B_0$. Because the variations of $B_0$ observed by MMS1 and MMS4 was slightly different from those observed by MMS2 and MMS3 (Supplementary Fig. 2c), which were separated from MMS1 and MMS4 in the direction perpendicular to $\mathbf{B}_0$ (Supplementary Fig. 1), such small-scale structures were probably dominated mainly by the direction perpendicular to $\mathbf{B}_0$. As discussed later, the background plasma velocity perpendicular to $\mathbf{B}_0$ was about 150 km s$^{-1}$ around 15:59:19.64 UT (Supplementary Fig. 2e). Because the maximum spacecraft separation was about 11 km, fluctuations of $B_0$ shorter than about 0.1 s may be caused by small-scale structures in the direction perpendicular to $\mathbf{B}_0$. Thus, we used 0.1-s moving averaged $B_0$ for the calculation of grad $B_0$ in FAC in addition to the original $B_0$ (Supplementary Fig. 2d). The same method was used to remove fluctuation for Event 2 (Supplementary Fig. 5d).

### Calculation of wave frequency

The calculation method of the wave frequency is the same as that used by ref. 42. A single rotation period was calculated from one right-handed rotation period of $\mathbf{B}_w$ in the FAC $x$-$y$ (perpendicular) plane. If

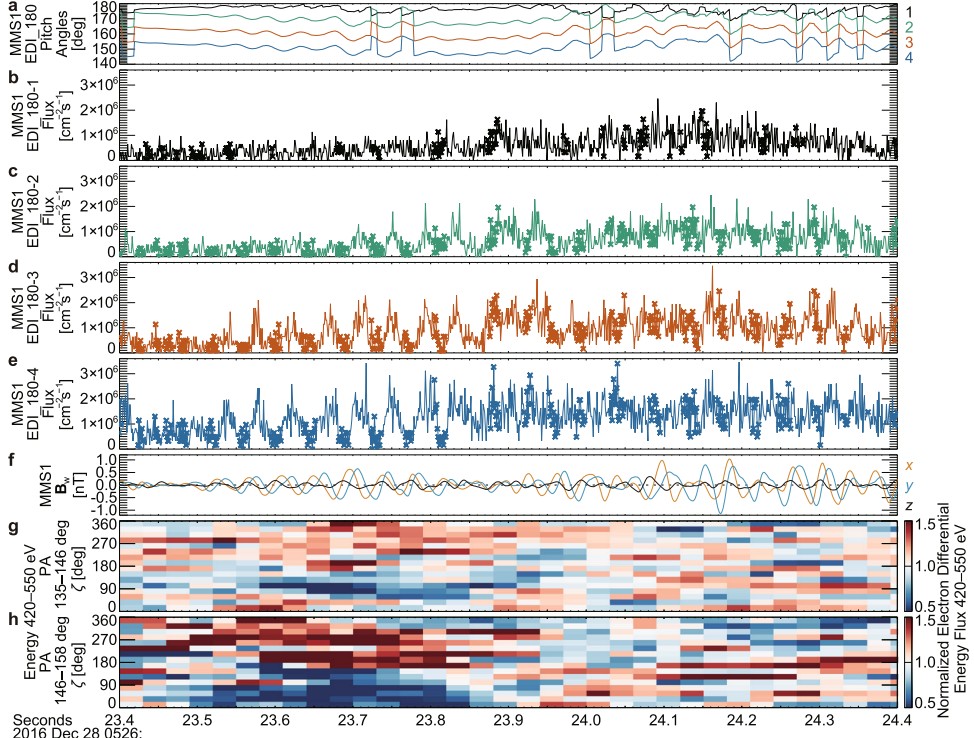

**Fig. 7 | Modulation of electron flux (Event 2).** The format is the same as that of Fig. 5. Both measurements by Fast Plasma Investigation-Dual Electron Spectrometers and EDI_180 indicate the existence of a dip of electron fluxes at $\zeta$ of about 90° around $V_{\mathrm{res}}$. When $B_{\mathrm{w}}$ was about 0.5 nT, electrons around 500 eV exhibited nongyrotropy in the pitch angle (PA) range of around 135°–165° (Supplementary Fig. 9d), this range is consistent with the width of trapping derived by a rough estimation.

the period of half a rotation before the observation time differed by a factor larger than two of that after the observation, the calculated period was rejected. If the amplitude of $\mathbf{B}_{\mathrm{w}}$ in the FAC $x$-$y$ plane became smaller than 0.01 nT during the single rotation or the ratio between the maximum amplitude and the minimum amplitude became larger than a factor of 1.5, the calculation of the period was stopped. We define the inverse of the period as the wave frequency.

**Analysis related to wave spectra (power, ellipticity, and Poynting flux angle)**

The methods to derive the wave power spectra of the magnetic field and the electric field (Fig. 1f, g), the ellipticity (Fig. 1h), and the Poynting flux angle (from $\mathbf{B}_0$) spectra (Fig. 1i) are the same as those used by ref. 42. The ellipticity and the angle of Poynting flux are plotted at the bins with the degree of polarization larger than 0.8. The methods to derive these parameters were originally proposed by refs. 60–62. The window length used for the analyses is 512 points (0.0625 s, 16-Hz frequency resolution) for Event 1 or 1024 points (0.125 s, 8-Hz frequency resolution) for Event 2 and 50% overlap. If a window includes multiple wave packets that have different characteristics, such analyses become less accurate[63,64]. Thus, we did not use these spectra for detailed analyses and mainly focused on the waveform.

**Electron and ion measurements by FPI**

Electrons in the energy range of about 6 eV–30 keV are measured by the FPI-DES[43,65]. We focused on electrons in the energy range of about 30–2000 eV, which included the energy range of the resonant current carriers (Fig. 3a). Because the spacecraft potential, which was measured by EDP with a temporal resolution of 8192 Hz (burst data), was only about 3–4 V (spacecraft are positively charged) around 15:59:19.6 UT (Event 1) or about 4–5 V around 05:26:23.7 UT (Event 2), we neglected the charging effect for the analysis. We neglected the relativistic effect in the data analysis, except for the estimate of $S$, because

the Lorentz factor $\gamma = 1/\sqrt{1 - v_e^2/c^2}$ for electrons with energies of <2 keV is <1.004, where $v_{\mathrm{e}}$ is the electron velocity.

The temporal resolution of the burst FPI-DES data (one full scan of VDF) is 30 ms (33.3 samples s$^{-1}$). This scan consists of 32 energy steps at 32 azimuthal angles and 16 elevation angles. The physical look directions of the spectrometers are approximated onto a regularized 11.25° elevation/azimuth grid. For this study, we reverse-corrected look directions to recover the irregular spaced look directions via the analyzer field of view intersected through a unit sphere[43,66]. The reverse-corrected VDFs were transformed to FAC. If the temporal resolution is much higher than the wave period[32], we can directly see the non-gyrotropic particle's VDFs rotating with the wave. However, it was not the case for whistler-mode waves here with a frequency higher than about 200 Hz (period of about 5 ms) for Event 1 or about 23 Hz (period of 43 ms) for Event 2. Thus, we disassembled the VDF data into individual energy/azimuth steps of measurements with a finer temporal resolution[32,35,36]. Eight sensor heads measure simultaneously per spacecraft, each of which simultaneously measures the 16 elevation angles via a segmented anode. Measurements are taken sequentially at each of 32 energy and 4 deflection (azimuth) step. This leads to 128 unique sample times, each with 128 simultaneous measurements, over a 30 ms energy/angle sweep. While the spacing between the times varies based on the step, the average spacing between measurements is about 234 μs, which is 30 ms divided by 128 steps. The integration time of each step is 196 μs, which is about 1/50 of the wave period in the present case. Although the time needed to switch to the next step depends on the energy and deflection state, the start times of each step are provided in the level-2 v3.4.0 FPI data. We calculated $\zeta$ for the center direction of the field of view of each step at the center of each measurement (integration time). For this calculation of $\zeta$, linearly interpolated $\mathbf{B}_{\mathrm{w}}$ in time was used. The disassembled data were binned by the pitch angle of 11.25° (32 bins) and by $\zeta$ of 22.5° (16 bins). To fill the bins as much as possible, we combined the data from all four MMS

spacecraft. For the analysis to see the temporal variation (Figs. 4 and 5), we combined the temporally nearest 3 VDFs from all four spacecraft (temporal resolution of about 0.1 s). For the analysis to include detailed features of the electron nongyrotropy (Fig. 3), 9 VDFs (270 ms) from each of four MMS spacecraft (total 36 VDFs) were used to get better statistics. Because the timings of measurements (time tags for VDFs) are not synchronized among spacecraft, we chose the combination of VDFs with the smallest maximum time difference. The time offsets from the four-spacecraft average are 7.560, 3.458, −5.267, and −5.750 ms for MMS1–4 for Event 1, respectively. Those for Event 2 are −5.908, 13.541, −1.401, and −6.232 ms for MMS1–4, respectively. Before the combination, the data from the four spacecraft were slightly corrected on the basis of the temporal average of electron pressure (average of diagonal components of the pressure tensor in the electron moment data) ratio (1.000, 1.033, 0.998, 0.980) (MMS1–4) over the entire 16-s interval (15:59:08–15:59:24 UT) for Event 1. The ratio for Event 2 (over the entire 8-s interval (05:26:21–05:26:29 UT)) was (1.000, 1.022, 0.997, 0.979). The PSD observed by each spacecraft was divided by this ratio. Note that such disassembled data are reliable only when the data are not lossy compressed[67,68]. The DES burst data were not lossy compressed in the interval shown in Fig. 4 (Event 1), although some of the data were lossy compressed just before this interval (15:59:16.9–15:59:18.2 UT). The DES burst data were not lossy compressed during the entire 8-s interval of Event 2 (05:26:21–05:26:29 UT).

We subtracted internal photoelectrons, which decrease steeply with increasing energy, from the electron VDF data[69]. Real electron fluxes were sufficiently large (a factor of ≥100(10) larger than the internal photoelectrons) in the energy range of about 50(30)–1,500 eV (16-s interval for Event 1) or about 50(30)–700(1,000) eV (8-s interval for Event 2) during most of the intervals. Although we did not permit the differential energy fluxes to become negative after the subtraction by setting negative values to 0, such a problem occurred only at high energies where the real fluxes were too small to contribute to currents.

The electron resonant current, which is rotating with the wave, was calculated as $\mathbf{J}_{res} = q n_e \mathbf{v}_{e\perp}$ in wFAC, where $n_e$ and $\mathbf{v}_{e\perp}$ are the density and the bulk velocity in the wFAC $x$-$y$ plane, respectively, as the moments of electrons. We calculated $\mathbf{J}_{res}$ using the electron VDF data in the energy range of 320.9–719.3 eV and the pitch angle range of 123.75°–157.5° for Event 1 (Fig. 3a) or in the energy range of 142.9–719.3 eV and the pitch angle range of 135.0°–168.75° for Event 2. Strongly nongyrotropic resonant electrons, which drove $\mathbf{J}_{res}$ ($J_x$ and $J_y$ in wFAC, which are perpendicular to $\mathbf{B}_0$), were included in the ranges (Figs. 4h, 6h). The resonant electrons with a smaller $v_{e\perp}$ ($=|\mathbf{v}_{e\perp}|$) (pitch angle closer to 180°) did not largely contribute to the current perpendicular to $\mathbf{B}_0$ because of small $v_{e\perp}$ and small $n_e$ due to a small volume in phase space. The PSD (and $n_e$) of resonant electrons with a larger $v_{e\perp}$ (higher energies) was too small to contribute to the current (Fig. 3a, Supplementary Fig. 9a).

Statistical errors for the histograms (Fig. 3b–d, Supplementary Fig. 9b–d) were estimated as $2\sqrt{\sum N}/\sum N$, where $N$ is the number of electron counts and $\Sigma$ means the total of counts in measurements that came from all four spacecraft. This statistical error was multiplied by the averaged differential energy flux to produce the 2σ error bars (corresponding to the 95% confidence interval). This approach is similar to the method used by ref. 33, although the calculation is simpler here, because one original measurement (pixel) from a single energy/deflection step always contributes to a single bin in the present case. We subtracted the contribution of the photoelectrons from $N$ using the ratio between the photoelectron flux and the residual (real) flux: the ratio was multiplied by the original $N$.

Ions in the energy range of about 2 eV–30 keV are measured by the FPI dual ion spectrometers (FPI-DIS)[43] with a temporal resolution of 0.15 s. Although the ion density from MMS1 was used in the main text, the

differences of ion and electron densities (Level-2 moments) among the spacecraft were within about 1% around 15:59:19.64 UT for Event 1 (Fig. 1c). For Event 2, although the electron densities were about 10% lower than ion densities, the calculated resonance velocities are not largely affected by the selection of spacecraft nor use of ion densities or electron densities (within 4% from $V_{res}$). Because the thermal velocity of ions is much smaller than that of electrons in the magnetosheath, the bulk velocity of ions is more reliable than that of electrons. Although the level-2 moments are calculated from the data above 10 eV, the contributions below 10 eV to the moments are expected to be small because of high ion temperature (parallel: about 500 (400) eV, perpendicular: about 700 (500) eV for Event 1 (Event 2)). The ion bulk velocities parallel ($v_{i\parallel}$) and perpendicular ($v_{i\perp}$) to $\mathbf{B}_0$ were about 160 and about 150 km s$^{-1}$, respectively, around 15:59:19.64 UT for Event 1 (Supplementary Fig. 2e). They were about 70 and about 50 km s$^{-1}$, respectively, around 05:26:23.69 UT for Event 2 (Supplementary Fig. 7e). Electron bulk velocities with the same magnitude were too small to be detected reliably, partly because of fluctuation caused by the whistler-mode wave. Because $v_{i\parallel}$ was much smaller than the phase velocity ($V_p = \omega/k$) (about 3,200 or about 800 km s$^{-1}$ for Event 1 and 2), the effect of Doppler shift for the wave was negligible. Because $v_{i\perp}$ was much smaller than $v_{e\perp}$ for resonant electrons in the energy and pitch angle ranges studied (>4,000 or >1,500 km s$^{-1}$ for Event 1 and 2) (Fig. 3a, Supplementary Fig. 9a), the effect of the drift motion perpendicular to $\mathbf{B}_0$ was also negligible for the electrons.

## Electron measurements by EDI

Although EDIs are primarily used for measurements of the electric field, they can also observe ambient electrons with fixed energy (500 eV in the present case) at very high temporal resolution[52]. During the interval of Event 1, EDIs only on MMS2 were in the ambient mode (amb-pm2). Each spacecraft has two EDIs on the opposite side of the spacecraft. Each EDI can look in any direction within a region greater than a 2π sr hemisphere. The sensor divides the annular area into 32 azimuthal sectors of 11.25° each, out of which 4 arbitrarily selectable channels can be selected. In the ambient mode (amb-pm2), electrons close to the pitch angle of 0° or 180° are continuously monitored by either of the EDIs. In addition, the next 3 channels in one of the directions on the sensor are recorded. Thus, under a favorable condition, electrons within about 40° from 0° or 180° (at a certain gyro phase) can be covered. For phenomena in which the temporal variation is much faster than the spin period of the spacecraft (20 s) or variation of the direction of $\mathbf{B}_0$ (Fig. 1a), whichever is faster, we can consider that the look direction of each channel was almost fixed. Because the period of the whistler-mode wave was only about 5 ms (Event 1) or about 43 ms (Event 2), the change of the look directions within several wave periods (almost equal to typical length of wave packets) was negligible. The temporal resolution of the burst data is about 1 ms (1,024 samples s$^{-1}$), which was about 1/5 or about 1/40 of the wave period.

## Validation of estimated $k$

Because the fluctuation of the parallel component ($z$ in FAC) of $\mathbf{B}_w$ was mostly much smaller than the other components (Fig. 5a), the assumption of parallel propagation is reasonable. We check the validity of $k$ derived under CPA. Because the derived wavelength ($\lambda = 2\pi/k$) of about 15 km is about 2.5 times larger than the maximum separation (about 6 km) of spacecraft in FAC $z$ (parallel to $\mathbf{B}_0$) for Event 1 (Supplementary Fig. 1), the phase difference of $\mathbf{B}_w$ between spacecraft is useful for the check. We use the pair of MMS1 and MMS4, which are close (about 2 km) in the direction perpendicular to $\mathbf{B}_0$. Around 15:59:19.64 UT, the instantaneous phase differences in the $x$-$y$ plane in FAC were distributed around 145°, which is expected by $k$ derived under CPA and the assumption of parallel propagation (Supplementary Fig. 2f).

**Table 2 | List of subfolders for MMS1 data at the Science Data Center**

| Instruments (datatype) | Subfolders at the science data center |
|---|---|
| FGM | data/mms1/fgm/brst/l2/2016/12/ |
| SCM | data/mms1/scm/brst/l2/scb/2016/12/ |
| EDP (waveform) | data/mms1/edp/brst/l2/dce/2016/12/ |
| EDP (spacecraft potential) | data/mms1/edp/brst/l2/scpot/2016/12/ |
| FPI-DES (moments) | data/mms1/fpi/brst/l2/des-moms/2016/12/ |
| FPI-DES (distribution functions) | data/mms1/fpi/brst/l2/des-dist/2016/12/ |
| FPI-DIS (moments) | data/mms1/fpi/brst/l2/dis-moms/2016/12/ |
| FPI-DIS (distribution functions) | data/mms1/fpi/brst/l2/dis-dist/2016/12/ |
| EDI | data/mms1/edi/brst/l2/amb-pm2/2016/12/ |
| Spacecraft position | data/mms1/mec/srvy/l2/epht89d/2016/12/ |

For MMS2, MMS3, or MMS4 spacecraft, mms1 in the URL should be replaced by mms2, mms3, or mms4.

For Event 2, the same pair of MMS1 and MMS4 was closest (about 3 km) in the direction perpendicular to $\mathbf{B}_0$. The derived $\lambda$ of about 34 km is about 4.3 times larger than the separation parallel to $\mathbf{B}_0$ (about 8 km). Around 05:26:23.69 UT, the instantaneous phase differences in the *x-y* plane in FAC were distributed around the expected value of 85° (Supplementary Fig. 7f).

## Data availability
MMS level-2 data analyzed in the present study are publicly available via the MMS Science Data Center (SDC) (https://lasp.colorado.edu/mms/sdc/public/). Data obtained by each instrument on MMS1 are in the subfolders listed in Table 2 at the SDC (mms1 in the URLs should be replaced by mms2, mms3, or mms4 for data obtained by other spacecraft). Source data required to generate the figures can be found at https://doi.org/10.5281/zenodo.7069800. The datasets generated during and/or analyzed in the present study are available from the corresponding author on reasonable request.

## Code availability
MMS data was loaded and analyzed using the Space Physics Environment Data Analysis System (SPEDAS)[70], which includes wavpol.pro that can derive wave power spectra, and spectra of the degree of polarization and ellipticity. SPEDAS is publicly available (http://themis.ssl.berkeley.edu/socware/bleeding_edge/). Ver. r30586 was used for the analysis. Ver. r31070 was used to make the final version of the plots with new color tables. Further detailed codes, which include a modified version of twavpol.pro and wavpol.pro in SPEDAS that can also derive spectra of the angle of Poynting flux from the background magnetic field, are available upon request to the corresponding author.

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

## Acknowledgements

This research was supported by the National Aeronautics and Space Administration (NASA) Magnetospheric Multiscale Mission (MMS) in association with NASA contract NNG04EB99C. Laboratoire de Physique des Plasmas (LPP) involvement for the search-coil magnetometer and Institut de Recherche en Astrophysique et Planétologie (IRAP) contributions to Fast Plasma Investigation (FPI) were supported by Centre National d'Études Spatiales (CNES) and Centre National de la Recherche Scientifique (CNRS). We acknowledge Thomas E. Moore, Jean-Andre Sauvaud, Conrad Schiff, John C. Dorelli, Levon A. Avanov, Benoit

Lavraud, Michael O. Chandler, and Victoria N. Coffey for providing instrumentation and data production/quality of FPI. We acknowledge Narges Ahmadi for the use of EDP data. We acknowledge Eric Grimes and the development team of the SPEDAS software for their fruitful efforts in providing this software. The following authors were supported by Grants-in-Aid for Scientific Research of Japan Society for the Promotion of Science supported N.K., T.A., Y.O., H.K, Y.S. (17H06140), Y.K. (18H03727), and M.K. (21K13979). This work is partially supported by Nagoya University Research Fund.

## Author contributions

N.K. identified the event, and performed the data analysis, interpretation, and manuscript preparation. T.A., Y.O., S.A.B., Y.M., M.K., Y.K., H.K., S.N., and M.S. contributed to data interpretation and the comparison to the theory. D.J.G., Y.S., S.Y., B.L.G., W.R.P., C.J.P., A.C.B., D.G.S., S.K. (FPI), O.L.C. (SCM), C.T.R., R.J.S (FGM), P.A.L., R.E.E., R.B.T. (EDP), and R.B.T. (EDI) contributed to the instrument development, operation, data processing, data quality assurance, and interpretation of the data. J.L.B. led the design and operation of the MMS mission and contributed to quality assurance and interpretation of the data. All authors reviewed the manuscript.

## Competing interests

The authors declare no competing interests.

## Additional information

[1]Institute for Space-Earth Environmental Research, Nagoya University, Nagoya, Japan. [2]Department of Earth and Planetary Science, Graduate School of Science, the University of Tokyo, Tokyo, Japan. [3]Research Institute for Sustainable Humanosphere, Kyoto University, Uji, Japan. [4]NASA Goddard Space Flight Center, Greenbelt, MD, USA. [5]Goddard Planetary Heliophysics Institute, University of Maryland, Baltimore County, MD, USA. [6]Department of Geophysics, Graduate school of Science, Tohoku University, Sendai, Japan. [7]Institute of Space and Astronautical Science, Japan Aerospace Exploration Agency, Sagamihara, Japan. [8]Department of Earth and Space Science, Graduate School of Science, Osaka University, Toyonaka, Japan. [9]Denali Scientific, Fairbanks, AK, USA. [10]Aurora Engineering, Potomac, MD, USA. [11]a.i. solutions Inc, Lanham, MD, USA. [12]Laboratoire de Physique des Plasmas, CNRS/Sorbonne Université/ Université Paris-Saclay/Observatoire de Paris/Ecole Polytechnique Institut Polytechnique de Paris, Paris, France. [13]Department of Earth, Planetary, and Space Science, University of California, Los Angeles, CA, USA. [14]Royal Institute of Technology, Stockholm, Sweden. [15]Laboratory for Atmospheric and Space Physics, University of Colorado, Boulder, CO, USA. [16]Department of Physics, University of New Hampshire, Durham, NH, USA. [17]Southwest Research Institute, San Antonio, TX, USA. ✉e-mail: kitamura@isee.nagoya-u.ac.jp

