## [Peer Review File · Nature Communications]

“Direct observations of energy transfer from resonant electrons to whistler-mode waves”, by N. Kitamura and coworkers, is a manuscript devoted to demonstrating that the non-gyrotropic distribution of electrons is a result of phase-trapping and has the same nonlinear dynamics as the case of whistler chorus. I find the paper well written, and the observations well analyzed, which makes the contents novel and of interest for a broad readership. However, I find that the conclusions are not so clearly supported by the evidence provided here. In the following, I explain my impression based on three remarks:

- (i) The non-gyrotropic distribution of electrons is frequently observed near reconnection by MMS as discussed, e.g., by Burtch et al. in *Science*, *Electron-scale measurements of magnetic reconnection in space*.
- (ii) On the other hand, the non-gyrotropic distribution of electrons is a topic of debate, where most quoted mechanisms are related to magnetic reconnection; e.g., *Effects of the guide field on electron distribution functions in the diffusion region of asymmetric reconnection*, by N. Bessho et al., *Physics of Plasmas*.
- (iii) The generation of whistler waves by these electrons is investigated by many studies. E.g., the most recent one by Choi et al., *Physics of Plasmas*, 2022: *Whistler waves generated by non gyrotropic and gyrotropic electron beams during asymmetric guide field reconnection*.

The current observation of non-gyrotropic distribution of electrons is related with magnetic reconnection and, therefore, could be caused by some other mechanisms as discussed in existing literature. Here, I am not arguing against the evidence that these electrons generate nearly parallel propagating whistler waves, which is fine and convincing.

I am also a bit surprised that Authors invoke well-known existing theory of non-linear wave growth when the wave have nearly constant frequency, while the typical case of chorus-like emission is characterized by frequency chirping. Meanwhile, Eq (2) is rather general and does not necessarily imply nonlinear growth. Again, I am not arguing against the well-established theory of non-linear wave growth; I am just emphasizing that its application to the present case may not be so obvious.

In summary, this work contains a very interesting analysis of MMS data. If the non-gyrotropic electron distribution was observed in the inner Earth's magnetosphere and at the same time of chorus events, then it would be a very strong evidence for the nonlinear wave-particle interaction theory and should be suitable for publication in *Nature Communications*. However, as noted above, the current observation of non-gyrotropic distribution of electrons is related with magnetic reconnection. Thus, in my opinion, attributing the non-gyrotropic distribution of electrons to phase-trapping and its generation of whistler waves by the same kind of process underlying chorus emission may be premature and controversial at this stage.

Reviewer #2 (Remarks to the Author):

Review of "Direct Observations of Energy Transfer from Resonant Electrons to Whistler-Mode Waves" by Kitamura, Amano, Omura, Boardsen et al.

The paper contains interesting new information, is well-written and should be published after a few corrections have been made.

Main Comments

The title is too general. This event occurred in a very small spatial region, that of the magnetosheath near a magnetic reconnection site. Non-gyrotropic electrons as shown in this paper may only exist at small sites like this one. I suggest that the authors add to the end of the title: "in the magnetosheath". There is no evidence that electromagnetic whistler mode waves detected in the magnetosphere or magnetosheath proper are due to resonance with non-gyrotropic electrons.

Whistler mode waves in the magnetosheath have been studied before and are called "lion roars" (JGR, 81, 13, 2261-2266, 1976; JGR, 87, A8, 6060-6072, 1982; JGR, 103, 4615-4626, 1998; AG, 17, 1528-1534, 1999). There is no evidence that in general lion roars are generated by non-gyrotropic electrons. Thus, this paper is showing a new generation mechanism and should be mentioned.

Many of the references given in the paper are not the discovery papers. This needs to be corrected throughout.

Minor Comments

Line 56, in reference to "pitch angle scattering" references, an early review of this topic is in RG, 35, 4, 491-502, 1997.

For electron acceleration, a fundamental reference is JGR, 110, A03225, 2005, doi:03210.01029/02004JA010811.

Line 58. For diffuse aurora a fundamental paper JGRSP, 120, 5943-5957, 2015.

Line 60. The first paper showing energetic electron anisotropy and electromagnetic waves was in Wave Inst. Spa. Plas., 55-62, Reidel Publ. Co., 1979. Another more recent article showing this nicely is JGR, 117, A11223, 2012. Doi:10.1029/2012JA018076.

Line 67. The first paper showing coherent electromagnetic waves was JGR, 114, A03207, doi:10.1029/2008JA013353, 2009. This should be added to the references.

Line 76. Why don't you use a more exact factor, $\sim 1,836$?

Line 77. It should be mentioned that ion cyclotron waves have been shown to be coherent: JGR, 120, 7536-7551, 2015. Doi:10.1002/2015JA021327.

Lines 82-83. This sentence is too general. Not all whistler mode waves are generated by nongyrotropic electrons. Please correct.

Line 246. The first paper to show coherent electromagnetic waves is JGR 114. A03207, 2009 doi:10.1029/2008JA013353. This should be referenced.

Reviewer #3 (Remarks to the Author):

Review of "Direct observations of energy transfer from resonant electrons to whistler-mode waves" by Kitamura et al.

This study shows strong nongyrotropy of cyclotron resonant electrons as direct evidence for the locally ongoing energy transfer from the resonant electrons to the whistler-mode waves using ultra-high temporal resolution data from the MMS mission. The methodology is mostly appropriate to support the conclusion and enough details are provided in the methods. Although the observation using ultra-high temporal resolution particle data is interesting and supports the nonlinear wave growth theory, I suggest the authors analyze more time intervals and do further numerical modeling to support the conclusion in a more comprehensive way, as described below.

Major suggestions:

Although the nonlinear growth calculation shows the positive growth rate, it is not adequate to lead to the conclusion that the observed particle distribution is indeed responsible for the observed whistler-mode wave properties. I suggest authors perform numerical modeling based on the observed plasma parameters to demonstrate that how the observed particle distribution leads to the observed wave properties (not only growth rates, but also other critical wave properties, such as wave normal angle, wave frequency spectra, etc.)

The growth rate calculation was only performed during one short period highlighted by gray vertical lines in Figure 1. However, whistler-mode waves with different wave spectra were observed in the prior period (15:59:10-15:59:19 UT) as well. I suggest the authors to analyze electron distributions, as well as perform similar growth rate calculation and numerical modeling in a few different periods to demonstrate the robustness of your results. The additional analyses could be moved to supplementary materials, if running out of space.

Minor suggestions:

For the observed whistler-mode waves in Figure 1, do they exhibit discrete structures (e.g., rising tones)? Please clarify it, preferentially with figures.

How is the total electron density achieved for the growth rate calculation (more specifically, V_{res} calculation)? Please clarify.

A few other minor suggestions:

L109–112: incomplete sentence, please rewrite.

In all color figures, the ticks in the colorbar don't show up. Please add them.

We are grateful to the reviewers for the comments. As indicated in the following responses, we have taken the comments and suggestions into account in the revised version of our manuscript as much as possible.

Analysis of an event in the magnetosheath was added to show that the nongyrotropy of cyclotron resonant electrons appears even without magnetic reconnection and the nongyrotropic resonant electrons provide energy to the whistler-mode wave.

The greatest achievement of this research is that we have shown by in-situ observations that electrons satisfying the cyclotron resonance condition exhibit (observable degree of) nongyrotropy and transfer energy to waves (expressed as $\mathbf{J}_{\text{res}} \cdot \mathbf{E}_w < 0$, which indicates secular energy transfer from the resonant electrons to the wave [Kato et al., 2013 (ref. 54)]). Without nongyrotropy, electrons cannot supply energy to the whistler-mode wave: for \mathbf{E}_w perpendicular to the magnetic field, $\mathbf{J}_{\text{res}} \cdot \mathbf{E}_w = 0$. The last paragraph of the main text was updated to emphasize this. Cyclotron resonant wave-particle interactions have been widely studied for many years, and it is naturally believed that whistler-mode waves are generated by cyclotron-resonant electrons in various cases in space. However, until now, no one has been able to observe the energy transfer rate ($\mathbf{J}_{\text{res}} \cdot \mathbf{E}_w$) in any region and case in space. Although the first example shown here happened to be near the magnetic reconnection site, we believe that it is extremely valuable because the general process of energy transfer from resonant electrons to whistler-mode waves in space was finally confirmed by in-situ observations. Furthermore, the ability to measure the gradient of magnetic field intensity made it possible to discuss even comparisons with nonlinear growth theory for the first time on the basis of observations. Although nongyrotropy of electrons rotating with whistler-mode waves contains many interesting features, we focused on the greatest achievement here as an initial report, and additionally performed some comparison with the nonlinear wave growth theory with measurable (local) parameters by in-situ multi-spacecraft measurements. We think that the other minor or detailed features should be discussed in another journal in the future. It is inevitable that spacecraft can only measure in-situ parameters, and arguments that strongly depend on parameters in regions that were not observed by the spacecraft are also beyond the scope of the present study. The outstanding point of the present study is the ability to demonstrate energy transfer from electrons to waves without the help of modeling that includes non-local parameters that cannot be validated by any in-situ observations.

During the re-check of calculations, we found that an incorrect value was used for the calculation of growth rate. The slight correction does not change any conclusion.

From the next page, responses to comments from each reviewer are described.
Comments from reviewers are displayed in blue.

Reviewer #1

“Direct observations of energy transfer from resonant electrons to whistler-mode waves”, by N. Kitamura and coworkers, is a manuscript devoted to demonstrating that the nongyrotropic distribution of electrons is a result of phase-trapping and has the same nonlinear dynamics as the case of whistler chorus. I find the paper well written, and the observations well analyzed, which makes the contents novel and of interest for a broad readership. However, I find that the conclusions are not so clearly supported by the evidence provided here. In the following, I explain my impression based on three remarks:

(i) The non-gyrotropic distribution of electrons is frequently observed near reconnection by MMS as discussed, e.g., by Burch et al. in *Science*, Electronscale measurements of magnetic reconnection in space.

(ii) On the other hand, the non-gyrotropic distribution of electrons is a topic of debate, where most quoted mechanisms are related to magnetic reconnection; e.g., Effects of the guide field on electron distribution functions in the diffusion region of asymmetric reconnection, by N. Bessho et al., *Physics of Plasmas*.

(iii) The generation of whistler waves by these electrons is investigated by many studies. E.g., the most recent one by Choi et al., *Physics of Plasmas*, 2022: Whistler waves generated by non gyrotropic and gyrotropic electron beams during asymmetric guide field reconnection.

The current observation of non-gyrotropic distribution of electrons is related with magnetic reconnection and, therefore, could be caused by some other mechanisms as discussed in existing literature. Here, I am not arguing against the evidence that these electrons generate nearly parallel propagating whistler waves, which is fine and convincing.

As pointed out, nongyrotropic electrons have been observed at the electron dissipation region (EDR) of magnetic reconnection [e.g., Burch et al., 2016 (ref. 51)]. However, the nongyrotropy for magnetic reconnection events is in a constant orientation with respect to the current layer and is essentially different from the nongyrotropy reported in the present study, which rotates with the frequency of the whistler-mode wave. This is added to the main text (line 197–200). Furthermore, the nongyrotropic electrons near the cyclotron resonance velocity came from the opposite side of the reconnection (in the opposite direction of wave propagation (Fig 1j)). This fact also supports the idea that they are not directly related to the nongyrotropic electrons near EDR. In addition, by showing another event in the magnetosheath, it was directly

shown that the same nongyrotropy of resonant electrons appears without magnetic reconnection and they provide energy to the whistler-mode wave.

We believe that what we have described above is sufficient for discussion in the main text. In addition, the entire electron distribution function exhibits nongyrotropy near EDR, while electrons became nongyrotropic only near the cyclotron resonance velocity during the wave events discussed in the present study. The location of the present event (Event 1) was downstream of the outflow region where ion jets had been observed for a long time, and we are not focusing on the vicinity of EDR. Thus, from various perspectives, the nongyrotropy is different from the nongyrotropy related to magnetic reconnection (especially EDR) and has no direct relation to it.

I am also a bit surprised that Authors invoke well-known existing theory of non-linear wave growth when the wave have nearly constant frequency, while the typical case of chorus-like emission is characterized by frequency chirping. Meanwhile, Eq (2) is rather general and does not necessarily imply nonlinear growth. Again, I am not arguing against the well-established theory of non-linear wave growth; I am just emphasizing that its application to the present case may not be so obvious.

Prior to Eq (2), the inhomogeneity factor (S) is obtained from the observed parameters. That is the part directly related to the discussion about the nonlinear wave growth. Although some resonant electrons are phase trapped under the condition of $|S| < 1$, no net energy transfer occurs when $S = 0$ [e.g., Omura et al. 2008 (ref. 13)]. For S to become an appropriate magnitude, either a frequency variation of the wave or a gradient of magnetic field intensity along the field line is necessary. For chorus waves in the magnetosphere, nonlinear wave growth is expected near the magnetic equator where the gradient of magnetic field intensity along the field line is almost zero. Thus, the frequency variation of the wave is necessary and the term for S related to the frequency variation is important. In contrast, the present observations demonstrate that a suitable magnitude of S for nonlinear wave growth is achieved by an appropriate magnitude of the gradient of the magnetic field intensity at the spacecraft position as opposed to the frequency variation seen in rising tones.

In summary, this work contains a very interesting analysis of MMS data. If the nongyrotropic electron distribution was observed in the inner Earth's magnetosphere

and at the same time of chorus events, then it would be a very strong evidence for the nonlinear wave-particle interaction theory and should be suitable for publication in Nature Communications. However, as noted above, the current observation of non-gyrotropic distribution of electrons is related with magnetic reconnection. Thus, in my opinion, attributing the non-gyrotropic distribution of electrons to phase-trapping and its generation of whistler waves by the same kind of process underlying chorus emission may be premature and controversial at this stage.

As described above, the nongyrotropy of electrons reported in the present study, which rotates with the frequency of the whistler-mode wave, essentially different from the nongyrotropy of electrons at EDR of magnetic reconnection, which has a constant orientation with respect to the current layer.

As pointed out, the chorus waves in the magnetosphere are the target that has been studied in the most detail regarding nonlinear wave growth. As mentioned in the main text, simulation studies have shown that nonlinear wave growth is caused by nongyrotropic resonant electrons [Hikishima and Omura, 2012 (ref. 53); Katoh et al., 2013 (ref. 54); Tao et al., 2017 (ref. 55); Hanzelka et al., 2021 (ref. 56); Nogi and Omura, 2021 (ref. 57)]. However, the high frequency of magnetospheric chorus wave (the order of kHz) requires higher temporal resolution for observations, and the small amplitude relative to the background magnetic field requires high pitch angle resolution. Furthermore, the small differential energy flux of the resonating electrons in the magnetosphere makes it difficult to obtain many electron counts. A quantitative estimate of these issues was made by Hanzelka et al. [2021 (ref. 56)], who concluded that it is difficult for existing instruments on spacecraft to capture the nongyrotropy associated with the chorus wave in the magnetosphere. However, if S is appropriate, the occurrence of phase trapping due to the wave is determined by mathematical calculations and is expected to be a general physical mechanism that does not depend on the region of space or the characteristics of plasma. Thus, we believe that we can state that this is the first directly observed example of nonlinear wave growth of whistler-mode waves that include chorus waves.

Reviewer #2

Review of “Direct Observations of Energy Transfer from Resonant Electrons to Whistler-Mode Waves” by Kitamura, Amano, Omura, Boardsen et al.

The paper contains interesting new information, is well-written and should be published after a few corrections have been made.

Main Comments

The title is too general. This event occurred in a very small spatial region, that of the magnetosheath near a magnetic reconnection site. Non-gyrotropic electrons as shown in this paper may only exist at small sites like this one. I suggest that the authors add to the end of the title: “in the magnetosheath”. There is no evidence that electromagnetic whistler mode waves detected in the magnetosphere or magnetosheath proper are due to resonance with non-gyrotropic electrons.

Cyclotron resonant wave-particle interactions have been widely studied for many years, and it is naturally believed that whistler-mode waves are generated by cyclotron-resonant electrons in various cases in space. However, until now, no one has been able to observe the energy transfer rate ($\mathbf{J}_{\text{res}} \cdot \mathbf{E}_w$) in any region and case in space. Although the examples shown in the present manuscript happened to be near the magnetic reconnection and in the magnetosheath, we believe that it is extremely valuable because the general process of energy transfer from resonant electrons to whistler-mode waves in space was finally confirmed by in-situ observations. Since the process is expected to be general, we have titled our manuscript general as well.

If the inhomogeneity factor (S) for nonlinear wave growth [e.g., Omura et al., 2008 (ref. 13)] is appropriate, the occurrence of phase trapping due to the wave is determined by mathematical calculations and is expected to be a general physical mechanism that does not depend on the region of space or the characteristics of plasma. As described in detail for the responses to Reviewer #1, Hanzelka et al. [2021 (ref. 56)] concluded that it is difficult for existing instruments on spacecraft to capture the nongyrotropy associated with chorus waves in the magnetosphere, although it is believed that a similar process occurs. To avoid misunderstanding that this is a phenomenon specific to magnetic reconnection, we have added an event in the magnetosheath. Although it is very important to show that similar phenomena occur in various regions and/or conditions in the future, we believe that it is a very important step to report the first case of such a phenomenon when it has not been previously observed at all.

Although more details, such as spatial scale, are a subject for the future, the spatial

scale of the growth region may be very limited part of the observed waves, if the waves grow efficiently in a short distance and propagate over a long distance after the end of the growth.

Whistler mode waves in the magnetosheath have been studied before and are called “lion roars” (JGR, 81, 13, 2261-2266, 1976; JGR, 87, A8, 6060-6072, 1982; JGR, 103, 4615-4626, 1998; AG, 17, 1528-1534, 1999). There is no evidence that in general lion roars are generated by non-gyrotropic electrons. Thus, this paper is showing a new generation mechanism and should be mentioned.

To the best of our knowledge, there is no indication that the lion roars are generated by nongyrotropic electrons. We added an event and briefly described the case of lion roars, which was unlikely related to magnetic reconnection, and additionally cited suggested papers there. Although it remains future work to determine whether similar generations occur commonly or not, we have shown that there are cases in which the lion roar is generated by nongyrotropic electrons by the first example.

Smith and Tsurutani [1976] <https://doi.org/10.1029/JA081i013p02261>

Tsurutani et al. [1982] <https://doi.org/10.1029/JA087iA08p06060>

Zhang et al. [1998] <https://doi.org/10.1029/97JA02519>

Baumjohann et al. [1999] <https://doi.org/10.1007/s00585-999-1528-9>

Many of the references given in the paper are not the discovery papers. This needs to be corrected throughout.

There is a huge amount of research related to the interaction of electrons and whistler-mode waves, and a large number of excellent papers. Because of the restriction on the number of references up to 70 in total, we needed to limit the citations to only papers that we felt were representative, rather than discovery papers, particularly on phenomena somewhat off the mainstream. We left a small (8) slot open because we expected that additional references may become necessary for revision. Within the limits, revisions were made to follow the suggestions as much as possible. If you think that some of them are still inappropriate, we may accept additional suggestions of replace if you provide information of the authors, title, and reasons why the suggested paper is better than originally cited paper.

Minor Comments

Line 56, in reference to “pitch angle scattering” references, an early review of this topic is in RG, 35, 4, 491-502, 1997.

Thank you very much for the suggestion. We have additionally cited the suggested paper [Tsurutani and Lakhina, 1997 (ref. 1)].

Tsurutani and Lakhina [1997] <https://doi.org/10.1029/97RG02200>

For electron acceleration, a fundamental reference is JGR, 110, A03225, 2005, doi:03210.01029/02004JA010811.

We appreciate the suggestion. We have cited the paper by Horne et al. [*Nature*, 2005 (ref. 11)] for the same purpose and think that it is not necessary to replace it, although the suggested paper [Horne et al., *JGR*, 2005] seems to be also one of important papers.

Horne et al. [2005] <https://doi.org/10.1029/2004JA010811>

Line 58. For diffuse aurora a fundamental paper JGRSP, 120, 5943-5957, 2015.

Thank you very much for the suggestion. The suggested paper [Hosokawa and Ogawa, 2015] focuses on ionospheric phenomena and does not show any whistler-mode waves. We think that the paper by Nishimura et al. [2010 (ref. 14)], which had already been cited and showed both of aurora and waves in space, must be better as a fundamental paper for diffuse aurora.

Hosokawa and Ogawa [2015] <https://doi.org/10.1002/2015JA021401>

Line 60. The first paper showing energetic electron anisotropy and electromagnetic waves was in *Wave Inst. Spa. Plas.*, 55-62, Reidel Publ. Co., 1979. Another more recent article showing this nicely is JGR, 117, A11223, 2012. Doi:10.1029/2012JA018076.

We could not find the first paper that you suggested. We think that the papers by Thorne & Tsurutani [1981 (ref. 22)] and Thorne et al. [2010 (ref. 15)], which had

already been cited, also clearly shown electron anisotropy and electromagnetic waves. We believe that they are the fundamental papers at early and relatively recent ages, respectively, and can play the almost same role as the suggested first and second [Kurita et al., 2012] papers.

Kurita et al. [2012] <https://doi.org/10.1029/2012JA018076>

Line 67. The first paper showing coherent electromagnetic waves was JGR, 114, A03207, doi:10.1029/2008JA013353, 2009. This should be added to the references.

We appreciate the suggestion. After considering it carefully again, it seems that there is no strict common understanding on what is considered coherent. Pursuing a rigorous definition of coherent does not change the conclusions of the present manuscript. We think that, for example, ‘near-monochromatic right-hand circularly polarized waves’ [Baumjohann et al., 1999 (ref. 27)] must be coherent. We additionally cited the suggested paper [Tsurutani et al., 2009 (ref. 28)] with the papers that include waveform and/or hodogram, which had indicated right-hand circularly polarization, of whistler-mode waves in the magnetosheath [Zhang et al., 1998 (ref. 26); Baumjohann et al., 1999 (ref. 27)].

Tsurutani et al. [2009] <https://doi.org/10.1029/2008JA013353>

Line 76. Why don't you use a more exact factor, $\sim 1,836$?

Although the cyclotron frequencies differ by a factor of 1836, only orders are shown, because cyclotron waves appear at various frequencies below the cyclotron frequencies.

Line 77. It should be mentioned that ion cyclotron waves have been shown to be coherent: JGR, 120, 7536-7551, 2015. Doi:10.1002/2015JA021327.

We believe that the details of EMIC waves themselves are beyond the scope of this manuscript. Thus, we only cite papers that had used similar analytical methods to those in the present manuscript for EMIC waves in the limit (70).

Remya et al. [2015] <https://doi.org/10.1029/2015JA021327>

Lines 82-83. This sentence is too general. Not all whistler mode waves are generated by nongyrotropic electrons. Please correct.

The sentence does not mean that all whistler-mode waves are generated by nongyrotropic electrons (line 85–86). If one can identify a strongly nongyrotropic electron velocity distribution function rotating with the whistler-mode wave around the cyclotron resonance velocity with a hole around ζ of $\sim 90^\circ$ (parallel propagating wave) or $\sim 270^\circ$ (anti-parallel propagating wave), it can be treated as smoking-gun evidence for locally ongoing energy supply to the wave.

Line 246. The first paper to show coherent electromagnetic waves is JGR 114. A03207, 2009 doi:10.1029/2008JA013353. This should be referenced.

The reference there [Omura, 2021 (ref. 21)] is not for ‘coherent waves’ [e.g., Tsurutani et al., 2009] but for ‘the nonlinear wave-particle interaction theory for coherent waves.’

Reviewer #3

Review of “Direct observations of energy transfer from resonant electrons to whistler-mode waves” by Kitamura et al.

This study shows strong nongyrotropy of cyclotron resonant electrons as direct evidence for the locally ongoing energy transfer from the resonant electrons to the whistler-mode waves using ultra-high temporal resolution data from the MMS mission. The methodology is mostly appropriate to support the conclusion and enough details are provided in the methods. Although the observation using ultra-high temporal resolution particle data is interesting and supports the nonlinear wave growth theory, I suggest the authors analyze more time intervals and do further numerical modeling to support the conclusion in a more comprehensive way, as described below.

Major suggestions:

Although the nonlinear growth calculation shows the positive growth rate, it is not adequate to lead to the conclusion that the observed particle distribution is indeed responsible for the observed whistler-mode wave properties. I suggest authors perform numerical modeling based on the observed plasma parameters to demonstrate that how the observed particle distribution leads to the observed wave properties (not only growth rates, but also other critical wave properties, such as wave normal angle, wave frequency spectra, etc.)

The growth rate calculation was only performed during one short period highlighted by gray vertical lines in Figure 1. However, whistler-mode waves with different wave spectra were observed in the prior period (15:59:10-15:59:19 UT) as well. I suggest the authors to analyze electron distributions, as well as perform similar growth rate calculation and numerical modeling in a few different periods to demonstrate the robustness of your results. The additional analyses could be moved to supplementary materials, if running out of space.

The heart of this study is not the growth rate but observations of the nongyrotropy of cyclotron resonant electrons. The fact that the resonant electron has nongyrotropy that rotates with the same frequency as the wave makes it possible to transfer energy for the wave, and the quantity that expresses this energy transfer is $\mathbf{J}_{\text{res}} \cdot \mathbf{E}_w$. For \mathbf{E}_w perpendicular to the magnetic field, nongyrotropy of electrons is necessary to make

nonzero $\mathbf{J}_{\text{res}} \cdot \mathbf{E}_w$. If electrons are completely gyrotropic, \mathbf{J}_{res} perpendicular to the background magnetic field becomes 0. Thus, it is crucial to show the existence and orientation (in the coordinate rotating with the wave) of the nongyrotropy. Since \mathbf{J}_{res} due to the nongyrotropy was almost antiparallel to \mathbf{E}_w , the error in the direction of \mathbf{J}_{res} has little effect on the inner product. It is obvious that \mathbf{J}_{res} in each energy-pitch angle bin is significant if the nongyrotropy is significant. The significance of the nongyrotropy has been shown by the histogram (Fig. 2b–d and supplementary Figure 9b–d). Since the data are independent in each energy-pitch angle bin, the fact that nongyrotropy shows a similar trend in multiple energy-pitch angle bins and becomes significant at the same time is very robust evidence of nongyrotropy. Furthermore, another independent observation by EDI at the same time supports the nongyrotropy identified by DES. Thus, without the need for more complex theory or modeling, we can conclude that the nongyrotropic electrons provided energy to the whistler-mode waves and contributed to their growth at the time. Such an observation of nongyrotropy is the ultimate direct evidence of wave-particle interactions (energy transfer) that can be observed in-situ by spacecraft.

Since the spatio-temporal scale of wave-particle interactions is unknown, it is not certain that the same interactions had been occurring at other times. There is little value in analyzing other time intervals with less clear nongyrotropy in detail. As shown in Figure 1j, because wave packets propagate along magnetic field lines, every wave packet can be observed by spacecraft only for an instant as it grows and propagates. The present study has shown that at least some of the wave packets had been still in the course of receiving energy from the resonant electrons. In other words, we have shown that the electrons can be significantly nongyrotropic to the extent that energy transfer rate becomes observable. By showing an example that such observation is possible, it is expected to trigger a variety of studies, including detailed studies of the occurrence frequency, temporal variation, spatial scale, and modeling.

Time intervals when the energy transfer from resonant electrons was not clearly visible means that the wave had been generated at the upstream (for the wave) region and likely had finished its effective growth. Because the magnetic field and electron distribution functions in the upstream generation region, which are critical to the wave properties, were not observable, it is not worthwhile to analyze such time intervals in detail. Even at the time when local energy transfer was occurring, the observed wave was the integrated result of wave-particle interactions from upstream to the location. We have shown that the inhomogeneity factor (S) became an appropriate magnitude for nonlinear growth at the spacecraft location due to the spatial gradient of the magnetic

field intensity. Because the spatial gradient is an essential factor, the simplification of spatial uniformity, which is often used when attempting modeling, is not possible. For the growth rate, the wave amplitude distributions and electron distribution functions at the downstream (for the wave) region also play a critical role, because the nongyrotopropy, which is directly related to the wave growth rate, is caused by the integration of wave-particle interactions at the downstream region, which is upstream for the resonant electrons. Thus, the information on the in-situ electromagnetic field and electron distribution function that can be observed by spacecraft are not sufficient to reproduce the observations of wave growth. Because simultaneous observations of the magnetic field intensity and electron distribution functions in all regions along the wave propagation from upstream to downstream are unrealistic and even the shape of the magnetic field line is not known precisely, modeling to reproduce the observed wave growth is extremely difficult. The primary result, though, is that the energy transfer rate at the spacecraft location can be shown by in-situ observations without such challenging modeling.

It would be useful to some extent to analyze another event where the nongyrotopropy (energy transfer) can be clearly identified. Since we added a new event to address the request from another reviewer, we discussed it briefly also.

Minor suggestions:

For the observed whistler-mode waves in Figure 1, do they exhibit discrete structures (e.g., rising tones)? Please clarify it, preferentially with figures.

No discrete structure is clearly visible in the wave spectrum. Since this is a short event, detailed spectral analysis is difficult. Thus, the temporal variation of the wave frequency obtained from the waveform is shown in Figure S2a. There is no particular trend other than frequency fluctuations at the boundaries of wave packets, where the phase of the wave is unstable and the wave period is difficult to determine accurately. For frequency and spectral variations on a long timescale of a few seconds or longer, the magnetic field intensity and electron distribution function at the spacecraft location changed. Thus, influence of spatial nonuniformity along the spacecraft pass (in the plasma rest frame) was expected to be significant and is beyond the scope of this study.

How is the total electron density achieved for the growth rate calculation (more

specifically, V_{res} calculation)? Please clarify.

Different from wave events in the magnetosphere, there is almost no cold plasma. The FPI level-2 moment data are used, which are described in Methods (lines 712–723). As an additional check, the phase difference between spacecraft was checked to confirm that the estimated wavelengths are correct in Methods (lines 752–766).

A few other minor suggestions:

L109–112: incomplete sentence, please rewrite.

We think that it is a complete sentence (lines 115–119).

In all color figures, the ticks in the colorbar don't show up. Please add them.

Thank you very much for the important suggestion. We had corrected all figures.

REVIEWER COMMENTS

Reviewer #1 (Remarks to the Author):

Authors have addressed all my prior remarks conscientiously. In particular, I have appreciated the careful explanation they provided about the fundamental difference of non-gyrotropic distribution of electrons discussed here with respect to that characteristic of EDR of magnetic reconnection.

On the other issue, connected with observation of non-gyrotropy associated with the chorus wave in the magnetosphere, I acknowledge that "it is difficult for existing instruments on spacecraft to capture" [56]. So, I accept that what the Authors propose here is an extrapolation on their observations based on an existing theory. It would be unfair to dismiss this work based on the request of an experimental evidence that will hopefully come in the future.

Based on this, I think that the manuscript can be accepted for publication in Nature Communications.

Reviewer #2 (Remarks to the Author):

Second Review of "Direct Observations of Energy Transfer from Resonant Electrons to Whistler-Mode Waves" by Kitamura, Amano, Omura, Boardsen, Gershman, Miyoshi, Kitahara, Katoh, Kojima, Nakamura, Shoji, Saito, Yokota, Giles, Paterson, Pollock, Barrie, Skeberdis, Kreisler, Le Contel, Russell, Strangeway, Lindqvist, Ergun, Torbert and Burch

The paper has been improved but only partially so. I reiterate some of my concerns from the first review.

The title is too general. The location of the event should be mentioned (in the magnetosheath).

The other referee had similar concerns that it is too early to assume that all whistler waves are generated in the same manner. For example the authors mention in response that lion roars may not be generated by nongyrotropic electrons. Lion roars are the major whistler mode waves in the magnetosheath. If the authors believe this, then a statement to this effect should be added to the paper.

The original findings of science results should always be cited. The article Wave Inst. Spa. Plas., 55-62, Reidel Publ. Co., 1979 is the original showing chorus and anisotropic energetic electrons. It should be cited as well.

Reviewer #3 (Remarks to the Author):

Second review of "Direct observations of energy transfer from resonant electrons to whistler-mode waves" by Kitamura et al.

I think the authors addressed my previous suggestions and concerns adequately. I only have a few minor suggested changes, as listed below. After revision, I recommend this paper to be published in Nature Communications.

L275: correspond -> corresponds; were -> was

L345: become -> becomes

L724: do not largely affected -> are not largely affected

Reviewer #1

Authors have addressed all my prior remarks conscientiously. In particular, I have appreciated the careful explanation they provided about the fundamental difference of non-gyrotropic distribution of electrons discussed here with respect to that characteristic of EDR of magnetic reconnection.

On the other issue, connected with observation of non-gyrotropy associated with the chorus wave in the magnetosphere, I acknowledge that "it is difficult for existing instruments on spacecraft to capture" [56]. So, I accept that what the Authors propose here is an extrapolation on their observations based an existing theory. It would be unfair to dismiss this work based on the request of an experimental evidence that will hopefully come in the future.

Based on this, I think that the manuscript can be accepted for publication in Nature Communications.

We are grateful for taking your time for this review.

Reviewer #2

Second Review of “Direct Observations of Energy Transfer from Resonant Electrons to Whistler-Mode Waves” by Kitamura, Amano, Omura, Boardsen, Gershman, Miyoshi, Kitahara, Katoh, Kojima, Nakamura, Shoji, Saito, Yokota, Giles, Paterson, Pollock, Barrie, Skeberdis, Kreisler, Le Contel, Russell, Strangeway, Lindqvist, Ergun, Torbert and Burch

The paper has been improved but only partially so. I reiterate some of my concerns from the first review.

We are grateful for taking your time for this review.

The title is too general. The location of the event should be mentioned (in the magnetosheath).

We add ‘in space’ to the title, and ‘in the magnetosheath’ to abstract. Although the examples shown in the present manuscript happened to be near the magnetic reconnection and in the magnetosheath, we believe that it is extremely valuable because the in-situ observations confirm the energy transfer from resonant electrons to whistler-mode waves, which must be general in space. Thus, we thought that it would be best to change the title to this one. For the more detailed region, we have also added the three more words ‘in the magnetosheath’ to the abstract, because we have room for three more words after the first revision. (Now the length of the abstract is at the upper limit.) The last sentence of the main text was replaced to describe the generality more appropriately (please see the end of the response for the next comment).

The other referee had similar concerns that it is too early to assume that all whistler waves are generated in the same manner. For example the authors mention in response that lion roars may not be generated by nongyrotropic electrons. Lion roars are the major whistler mode waves in the magnetosheath. If the authors believe this, then a statement to this effect should be added to the paper.

We have not intended to state that all whistler-mode waves are generated in the same manner from the beginning. Use of articles in a sentence was corrected (lines 81–84). (We had not noticed that it causes such a misunderstanding until this revision.)

An additional misunderstanding may be caused by a wrong tense in the previous response. There has not been no indication until the present study that the lion roars are

generated by nongyrotropic electrons. The present results have shown that there are cases in which the lion roars are generated by nongyrotropic electrons. Especially, Event 2 is one of the typical lion roars in the magnetosheath. Thus, at least some of the lion roars are generated by nongyrotropic resonant electrons, although whether all or some of them are generated in the same manner will be a subject for future study. To be clear that we do not intend to conclude that all whistler waves are generated in the same manner, the last sentence of the main text was replaced by the following sentences.

Although the nonlinear wave growth due to phase trapping of electrons has been discussed exclusively for whistler-mode waves in the magnetosphere, identification of nongyrotropy has not been established there. The successful identification near the reconnection and in the magnetosheath indicates that the nonlinear wave growth may play a role in broader applications in space if the appropriate condition is satisfied.

The original findings of science results should always be cited. The article Wave Inst. Spa. Plas., 55-62, Reidel Publ. Co., 1979 is the original showing chorus and anisotropic energetic electrons. It should be cited as well.

We agree that the original findings of science results should be cited. We could not find the book until the last revision, because the title of the book was abbreviated and information of the authors nor the title were provided. We continued the search, and found the manuscript [Tsurutani et al., 1979]. Since the number of cited papers has already reached the limit (70), we decided to remove the manuscript by Li et al. [2014] from the citation and selected one to be added. Because of various limitations of the old observations, it was difficult to select one manuscript that should be regarded as the original showing chorus and anisotropic energetic electrons. Among several candidates, we selected the manuscript by Anderson and Maeda [1977] that was cited by Tsurutani et al. [1979], because it was earlier and electron anisotropy in a wider energy range was analyzed.

Reviewer #3

Second review of “Direct observations of energy transfer from resonant electrons to whistler-mode waves” by Kitamura et al.

I think the authors addressed my previous suggestions and concerns adequately. I only have a few minor suggested changes, as listed below. After revision, I recommend this paper to be published in Nature Communications.

L275: correspond -> corresponds; were -> was

L345: become -> becomes

L724: do not largely affected -> are not largely affected

We are grateful for taking your time for this review. We corrected these parts as suggested.

REVIEWERS' COMMENTS

Reviewer #2 (Remarks to the Author):

The very nice paper is acceptable for publication in Nature Communications.